# Protein Arginylation Is Regulated during SARS-CoV-2 Infection

**DOI:** 10.3390/v15020290

**Published:** 2023-01-19

**Authors:** Janaina Macedo-da-Silva, Livia Rosa-Fernandes, Vinicius de Morais Gomes, Veronica Feijoli Santiago, Deivid Martins Santos, Catarina Maria Stanischesk Molnar, Bruno Rafael Barboza, Edmarcia Elisa de Souza, Rodolfo Ferreira Marques, Silvia Beatriz Boscardin, Edison Luiz Durigon, Claudio Romero Farias Marinho, Carsten Wrenger, Suely Kazue Nagahashi Marie, Giuseppe Palmisano

**Affiliations:** 1GlycoProteomics Laboratory, Department of Parasitology, ICB, University of São Paulo, São Paulo 05508-000, Brazil; 2Laboratory of Experimental Immunoparasitology, Department of Parasitology, ICB, University of São Paulo, São Paulo 05508-000, Brazil; 3Unit for Drug Discovery, Department of Parasitology, Institute of Biomedical Sciences at the University of São Paulo, São Paulo 05508-000, Brazil; 4Laboratory of Antigen Targeting for Dendritic Cells, Department of Parasitology, Institute of Biomedical Sciences at the University of São Paulo, São Paulo 05508-000, Brazil; 5Laboratory of Clinical and Molecular Virology, Department of Microbiology, ICB, University of São Paulo, São Paulo 05508-000, Brazil; 6Laboratory of Molecular and Cellular Biology (LIM 15), Department of Neurology, Faculdade de Medicina FMUSP, Universidade de São Paulo, São Paulo 01246-903, Brazil; 7School of Natural Sciences, Macquarie University, Sydney 2109, Australia

**Keywords:** arginylation, SARS-CoV-2, COVID-19, N-degron pathway, viral infection

## Abstract

Background: In 2019, the world witnessed the onset of an unprecedented pandemic. By February 2022, the infection by SARS-CoV-2 has already been responsible for the death of more than 5 million people worldwide. Recently, we and other groups discovered that SARS-CoV-2 infection induces ER stress and activation of the unfolded protein response (UPR) pathway. Degradation of misfolded/unfolded proteins is an essential element of proteostasis and occurs mainly in lysosomes or proteasomes. The N-terminal arginylation of proteins is characterized as an inducer of ubiquitination and proteasomal degradation by the N-degron pathway. Results: The role of protein arginylation during SARS-CoV-2 infection was elucidated. Protein arginylation was studied in Vero CCL-81, macrophage-like THP1, and Calu-3 cells infected at different times. A reanalysis of in vivo and in vitro public omics data combined with immunoblotting was performed to measure levels of arginyl-tRNA-protein transferase (ATE1) and its substrates. Dysregulation of the N-degron pathway was specifically identified during coronavirus infections compared to other respiratory viruses. We demonstrated that during SARS-CoV-2 infection, there is an increase in ATE1 expression in Calu-3 and Vero CCL-81 cells. On the other hand, infected macrophages showed no enzyme regulation. ATE1 and protein arginylation was variant-dependent, as shown using P1 and P2 viral variants and HEK 293T cells transfection with the spike protein and receptor-binding domains (RBD). In addition, we report that ATE1 inhibitors, tannic acid and merbromine (MER) reduce viral load. This finding was confirmed in ATE1-silenced cells. Conclusions: We demonstrate that ATE1 is increased during SARS-CoV-2 infection and its inhibition has potential therapeutic value.

## 1. Introduction

In 2019, the world witnessed the onset of an unprecedented pandemic [1]. Patients in the capital and largest city in China’s Hubei province, Wuhan, developed pneumonia associated with infection with a new type of coronavirus called severe acute respiratory syndrome coronavirus 2 (SARS-CoV-2) [2,3]. The clinical symptoms presented by infected patients ranged from mild to severe and included nonspecific manifestations such as fever, cough, sore throat, respiratory failure, muscle damage, and death [2,4,5,6]. Although there had been a worldwide mobilization to contain the exponential spread of SARS-CoV-2 [7,8], in February 2022, the deaths of more than 5 million people in the world were due to the new coronavirus infection [9] (https://covid19.who.int/), accessed on 1 February 2022. The search for effective treatments and measures to fight the pandemic has driven studies that seek to elucidate the infectious mechanisms of SARS-CoV-2 [10]. As it belongs to the Coronaviridae family, the new coronavirus has similarities in structure and pathogenicity with SARS-CoV, both being single-stranded positive RNA viruses (+ssRNA) [11]. However, differences in the structural Spike (S) glycoprotein were identified in SARS-CoV-2, which partly contributed to greater efficiency in its dissemination than other coronaviruses [12,13].

During the replication cycle, the SARS-CoV proteins use the host’s endoplasmic reticulum (ER) to induce the formation of double-membrane vesicles for RNA synthesis, followed by the assembly of virions in the ER-Golgi intermediate compartment by structural proteins [14,15]. The intense use of the host’s ER during viral replication increases the stress in this compartment, resulting in the accumulation of misfolded proteins and activation of the unfolded protein response (UPR) [15,16]. Recently, we and other groups discovered that SARS-CoV-2 infection also induces ER-stress and may be associated with patient survival [17,18,19]. The UPR pathway is highly conserved and regulates important cellular events such as growth, defense, homeostasis, and cell survival [13,20]. In addition, activation of this pathway can also stimulate tissue repair processes. Degradation of misfolded/unfolded proteins is an essential element of proteostasis [21] and occurs mainly in lysosomes or proteasomes, which degrade long- and short-lived proteins, respectively [22,23,24].

Ubiquitination is a universal tagging for protein degradation and is recognized by the proteasome [25,26]. The N-terminal arginylation of proteins is characterized as an inducer of ubiquitination and proteasomal degradation by the N-degron pathway, as elucidated by Varshavsky et al. [27,28]. A direct relationship between the half-life of a protein and its N-terminal residue has been demonstrated [27,28]. While methionine promotes protein stability, other charged or bulky amino acids, including arginine, result in rapid degradation [27,28]. Enzymes that remove the N-terminal methionine from nascent proteins, such as METAP1 and METAP2, can expose residues to arginylation from the neo-N-terminus. Studies have demonstrated arginylation of different proteins in oxidized cysteine residues, or aspartic acid and glutamic acid exposed at the N-terminus [29,30,31,32,33]. Asparagine and glutamine are tertiary destabilizing residues, as they can undergo an enzymatic deamidation reaction and be converted into glutamic acid and aspartic acid, recognized as secondary sites [33]. Thus, arginylated proteins or protein fragments with tertiary and secondary residues exposed at the N-terminus are recognized by N-recognins, ubiquitinated, and degraded by the proteasome-pathway [34,35]. Moreover, internal aspartic and glutamic acids are specifically arginylated on several proteins involved in different biological processes [32,36,37].

Protein arginylation has been associated with cellular stress conditions [38], including ER stress, oxidative stress, and misfolded protein stress [38,39,40]. Under these conditions, arginylation protects against cell death [40]. It was demonstrated that the viral proteins S, ORF3a, ORF6, ORF7a, ORF8ab, and ORF8b of SARS-CoV are able to induce ER stress. Moreover, the use of pharmacological inhibitors of the UPR pathway regulates viral replication [41]. Upregulation of this pathway has been identified also in cells infected with SARS-CoV-2 and murine hepatitis virus (MHV), often used as a model for the Betacoronavirus genus. Infection with only the viral proteins S and ORF8 induced the activation of the UPR, revealing the importance of this pathway for the success of the infection [42]. Furthermore, the activity of arginyl-tRNA-protein transferase (ATE1), the enzyme that promotes arginylation, is necessary to decrease mutation events when a cell is subjected to stressful conditions that damage DNA [39]. In 2002, the knockout of the ATE1 gene resulted in abnormalities in important processes such as cardiac development, angiogenesis, and tissue morphogenesis in mammals [43]. However, studies of the role of arginylation in infectious diseases are still scarce [44,45,46,47,48,49]. Wang et al. 2017 reported that infection by a picorna-like virus in infected Drosophila cells results in suppression of the N-degron rule pathway and inhibits apoptosis, benefiting viral replication. In addition, arginylated peptides have already been identified in trypanosomatids and the putative protein ATE1 has been identified in *P. falciparum*, the etiologic agent of malaria. The enzyme has been shown to have a prokaryotic-like sequence, however, eukaryotic transferase specificity [50]. Modulation of the N-degron pathway has been shown to influence *Bacillus anthracis* infection, as the EF adenylate cyclase toxin is a substrate of the pathway and is critical for the progression of pathophysiology.

In this manuscript, we elucidate the role of arginylation during SARS-CoV-2 infection. We conducted a study on modulation of the N-degron pathway and protein arginylation in Vero CCL-81, macrophage-like THP1, and Calu-3 cells infected at different times. A reanalysis of public omics data combined with Western blotting was performed to measure the levels of ATE1 and arginylated proteins. Dysregulation of the N-degron pathway was specifically identified during coronavirus infection, not in other respiratory viruses. We demonstrated that during SARS-CoV-2 infection there is an increase in the expression of the ATE1 enzyme and its modified substrates in Calu-3, HEK 293T, and Vero CCL-81 cells. On the other hand, infected macrophages showed no ATE1 regulation. Moreover, ATE1 modulation was found to depend on the SARS-CoV-2 variant, as shown with P1 and P2 viral infections and transfection with the spike glycoprotein receptor-binding domain (RBD) of different variants. In addition, ATE1 inhibitors, tannic acid and merbromine (MER) reduced viral load, and this was confirmed in ATE1 silenced cells.

## 2. Methods

### 2.1. Data Sources and Curation

Previously published studies were used to verify the abundance of proteins that make up the N-degron pathway in the non-infected and SARS-CoV-2 infected groups: (i) Saccon et al. [51] (Calu-3, Caco-2, Huh7, and 293FT cell lines, proteomics); (ii) Nie et al. [52] (autopsy 7 organs, 19 patients, proteomics); (iii) Leng et al. [53] (lung tissue, 2 patients, proteomics); (iv) Qiu et al. [54] (lung tissue, 3 patients, proteomics); (v) Bojkova et al. [55] (Caco-2 cells, proteomics); (vi) Wu et al. [56] (lung tissue, colonic transcriptomics); and (vii) Desai et al. [57] (lung tissue, transcriptomics). To verify modulation of the N-degron pathway in other viral infections, including MERS-CoV/SARS-CoV/H1N1 influenza virus/Respiratory syncytial virus (RSV), data from the following studies were evaluated: (viii) Zhuravlev et al. [58] (MRC-5, A549, HEK293FT, and WI-38 VA-13 cell lines, H1N1 influenza virus, transcriptomics); (ix) Li et al. [59] (A549 and 293T cell lines, H1N1 influenza virus, transcriptomics); (x) Krishnamoorthy et al. [60] (comparative among coronaviruses, transcriptomics); (xi) Ampuero et al. [61] (time course of RSV infection in the lung, transcriptomics); (xii) Besteman et al. [62] (RSV infected neutrophils, transcriptomics); and (xiii) Dave et al. [63] (RSV infected alveolar cell, proteomics). Deep proteome data from non-infected cell lineages were recently made available publicly by Zecha et al. [64] to model SARS-CoV-2 infection in Vero E6 (kidney epithelial cell, African green monkey), Calu-3 (lung adenocarcinoma), Caco-2 (colorectal adenocarcinoma), and ACE2-A549 (lung carcinoma expressing ACE2 to gain cellular entry). The iBAQ intensities of proteins that make up the N-degron pathway were evaluated without infection to access the basal levels of arginylation-related proteins. Experimentally arginylated proteins were retrieved from the datasets of Seo et al. [65] and Wong et al. [66] to access the regulation levels of these proteins during SARS-CoV-2 infection. Single-cell RNA-seq data from nasopharyngeal samples provided by Chua et al. [67] were reanalyzed to identify cell clusters expressing the ATE1 enzyme.

### 2.2. Bioinformatics Analysis

The tidyverse [68], biostrings, and seqinr [69] packages were used to map potentially arginylated proteins in the *Homo sapiens* and *Chlorocebus sabaeus* proteomes (downloaded in 1 May 2021, https://www.uniprot.org/, accessed on 1 May 2021). Signal peptide sequences were removed. Only proteins that have the potential to be arginylated at the N-terminus (NtE, NtD, NtC, NtN, NtQ) were retained. Caspase-generated fragments were not considered. It is important to mention that this list contains potential arginylated proteins that need to be confirmed experimentally and other targets might not be present in this list. The corrplot package was used to evaluate the correlation between proteins/genes, applying a Spearman test with a cut-off significance of *p*-value < 0.05. Protein subcellular locations were determined by UniProt release 12.4 (https://www.uniprot.org/news/2007/10/23/release, accessed on 1 May 2021) and the pRoloc package [70]. The analysis of gene ontology (GO) was determined by the g:profile [71] and DAVID [72] tools. A q-value threshold of 0.05 was used, corrected by the Benjamini–Hochberg method [73]. InteractiVenn was used to build the Venn diagrams [74]. The String database v.11.5 was applied for protein network analysis (https://string-db.org/, accessed on 1 May 2021) with the following parameters: medium confidence score (0.400), text mining, coexpression, and neighborhood enabled.

### 2.3. Single-Cell RNA-seq Re-Analysis

Expression matrices were loaded into RStudio (v. 4.0.3) with the Seurat package [75]. A filter to remove cells with less than 200 expressed genes or more than 25% of mitochondrial transcripts was applied using the ‘subset()’ function in each sample. Then, cell counts were log-normalized by a size factor of 10,000 RNA counts and feature selection was performed by selecting the 2000 genes with the highest dispersion. Unsupervised identification of anchor correspondences between the canonical correlation analysis (CCA) space of each sample’ normalized data was performed with the ‘FindIntegrationAnchors()’ function with 30 dimensions. After that, the data were integrated by ‘IntegrateData()’ function and scaled using ‘ScaleData()’. Principal component analyses (PCA) and uniform approximation and projection dimension reduction (UMAP) with 30 principal components were applied. A nearest neighbor plot using 30 PCA reduction dimensions was calculated using ‘FindNeighbors()’, followed by clustering using ‘FindClusters()’ with a resolution of 0.5. The Metaboanalyst platform [76] was used to evaluate differently regulated genes between cell clusters identified in the single-cell RNA-seq analysis.

### 2.4. Cell Culture

Vero CCL-81 cells were cultured in DMEM medium supplemented with 10% fetal bovine serum (FBS), 100 U/mL penicillin-streptomycin, 4.5 g/L glucose, 2 mM L-glutamine, 1 mM sodium pyruvate, and 1.5 g/L NaHCO_3_. Calu-3 cells were cultured in DMEM medium supplemented with 20% FBS, 1% non-essential amino acids, 4.5 g/L glucose, 2 mM L -glutamine, 1 mM sodium pyruvate, 100 U/mL penicillin-streptomycin and 1.5 g/L NaHCO_3_. THP-1 cells were cultured in RPMI-1640 supplemented with 10% FBS, and 1% penicillin-streptomycin at 37 °C. All cells were kept in a humidified 5% CO_2_ atmosphere. THP-1 monocyte cells were differentiated into macrophage-like cells as described by Gatto et al. [77] with few modifications. THP-1 monocytes were induced to differentiate into macrophages by the addition of phorbol-12-myristate 13-acetate (PMA, 50 ng/mL) (ab120297, Abcam, Cambridge, UK) for 48 h (h). After this time, the PMA-containing medium was replaced with fresh medium without PMA for 24 h prior to SARS-CoV-2 infection. Cell differentiation was verified by evaluating cell adhesion and spreading under an optical microscope.

### 2.5. Viral Infection

In this study, SARS-CoV-2 isolate HIAE-02: SARS-CoV-2/SP02/human/2020/BRA (GenBank accession number MT126808) [78], P1 variant (IMT03—Man 87209, Gsaid EPI_ISL_1060981), and P2 variant P2 (LMM—38135, Gsaid EPI_ISL_770561) were used in all infections with multiplicity of infection (MOI) of 0.02. Following adsorption in DMEM with 2.5% FBS for 1 h, fresh medium was added, and the cells were further incubated at 37 °C and 5% CO_2_ for the time described in each experiment. All experiments were performed with cell controls that underwent the same process in the absence of virus, named mock-infected cells.

All assays were performed in biological triplicates in a BSL-3 facility at the Institute of Biomedical Sciences, University of Sao Paulo, under the Laboratory biosafety guidance related to coronavirus disease (COVID-19): Interim guidance, 28 January 2021 (https://www.who.int/publications/i/item/WHO-WPE-GIH-2021.1, accessed on 1 May 2021).

### 2.6. Time-Course Evaluation of Protein Arginylation during Viral Infection

For comprehensive time course evaluation, Vero CCL-81 and Calu-3 cells were infected with SARS-CoV-2. Cell lysates were collected at 2, 6, 12, 24, and 48 h post infection (hpi) in 8 M urea supplemented with protease (cOmplete, Sigma-Aldrich, St. Louis, MO, USA) and phosphatase inhibitors (PhosStop, Sigma-Aldrich). Aliquots of cells and supernatants were collected at the different time points for virus RNA copy numbers quantification by reverse transcription-quantitative polymerase chain reaction (RT-qPCR), targeting the E gene as previously described [79].

### 2.7. Chemical Inhibition of Protein Arginylation

To evaluate the effects of protein arginylation inhibition, Calu-3 cells and differentiated macrophages were incubated with 25 µM of merbromin (Mercury dibromofluorescein disodium salt, Sigma-Aldrich), 1 µM of tannic Acid (Sigma-Aldrich) or medium for 1 h prior to infection with SARS-CoV-2. Following adsorption in DMEM with 2.5% FBS for 1 h, infected and respective mock infected cells were kept at the same inhibitor concentration for 24 and 72 hpi at 37 °C and 5% CO_2_. Cell lysates were collected in BE buffer (HEPES 10 mM, SDS 1%, MgCl2.6H2O 1.5 mM, KCl 10 mM, DTT 1 mM, NP-40 0.1%) containing protease (cOmplete, Sigma-Aldrich) and phosphatase inhibitors (PhosStop, Sigma-Aldrich).

### 2.8. siRNA-Directed Inhibition of ATE1

Predesigned siRNAs for the ATE1 transcript (siATE1) (hs.Ri.ATE1.13.3) were purchased from Integrated DNA Technologies (IDT, Coralville, IA). Calu-3 cells were transfected with 3 µL Lipofectamine 3000 reagent (Thermo Fischer Scientific, Waltham, MA, USA) alone (control) or with 30 pmol of siRNA-*ATE1* in 12 well plates, according to the manufacturer’s recommendation. After incubation for 2 h at 37 °C and 5% CO_2_, fresh cell culture medium supplemented with 5% FBS was added to each condition. At 48 h after transfection, viral and mock infections were carried out as described above. Cell lysates were collected in BE buffer containing protease (cOmplete, Sigma-Aldrich) and phosphatase inhibitors (PhosStop, Sigma-Aldrich).

### 2.9. Viral Quantification

Aliquots of supernatants from infected or mock-infected cells undergoing the above-mentioned treatments were collected at the different conditions for RNA extraction using TRIzol reagent (ThermoFisher, Waltham, MA, USA) according to the manufacturer’s instructions. Viral copy number quantification by RT-qPCR was performed using Detection Kit for 2019 Novel Coronavirus (2019-nCoV) RNA (PCR-Fluorescent Probing) (Cat. #DA-930) (China) in a QuantStudio 3 real-time PCR system (Applied Biosystems, Waltham, MA, USA) according to the manufacturer’s instructions. The percentage of viral release was calculated with the CTs values of the experimental triplicates. Graphics were created using GraphPad Prism software version 8.1 (GraphPad Software, San Diego, CA, USA).

### 2.10. Transient Transfection of HEK 293T Cells

Six hundred thousand human embryonic kidney (HEK) 293T cells (ATCC No CRL-11268) were cultured in 6-well plates (TPP) in Dulbecco’s Modified Eagle’s Medium (DMEM; Life Technologies, Carlsbad, CA, USA), supplemented with 10% ultra-low heat inactivated fetal bovine serum (FBS, Life Technologies), 1x l-glutamine and 1x antibiotic-antimycotic (all from Life Technologies). The following day, FBS-supplemented DMEM was washed off and replaced by 2 mL fresh medium prior to polyethyleneimine (PEI)-mediated transfection with a plasmid expressing either full-length spike protein (kindly provided by Dr. Jason S. McLellan, The University of Texas [80]) or with plasmids encoding the Spike protein receptor binding domain (RBD) amino acids 319 to 541 from the Wuhan strain (available at BEI Resources #NR-52309, https://www.beiresources.org/Catalog/BEIPlasmid Vectors/NR-52309.aspx, accessed on 1 May 2021) and from the beta, gamma (P1), and delta (synthesized by Genscript, Piscataway, NJ, USA). An empty vector was used as the control. One μg of each vector DNA was added to a final volume of 100 μL 150 mM NaCl solution containing 0.45 μg of PEI per μg of DNA. The mix was vortexed for 10 s, incubated for 10 min at room temperature, and evenly distributed in each well. Culture supernatants were removed 24, 48, 96, and 120 h after transfection, and 300 μL of BE buffer (HEPES 10 mM, SDS 1%, MgCl_2_.6H_2_O 1.5 mM, KCl 10 mM, DTT 1 mM, NP-40 0.1%) were added to each well to lyse the cells. The cell lysate was transferred to a 500 μL Eppendorf tube and frozen at −20 °C until use.

### 2.11. Western Blot

Proteins were extracted from cellular lysates and quantified using the Qubit Protein Assay Kit platform (Invitrogen) according to the manufacturer’s instructions. A total of 15 µg of proteins were separated by SDS-PAGE and electro-transferred to PVDF membranes, which were directly incubated with blocking buffer (5% bovine serum albumin (BSA) in Tris-buffered saline (TBS) at 0.05% Tween-20 (TBST) for 1 h. Subsequently, the samples were incubated overnight with primary antibodies (Table 1) and washed three times with TBST. Then, the bands were incubated with the respective secondary antibodies for 1 h at room temperature. Immunoreactive bands were detected with the ChemiDoc XRS Imaging System equipment and protein quantification was performed using the ImageJ software. Graphs were plotted using GraphPad Prism version 8.1 software. Bands with statistically significant intensities among groups were evaluated by applying an Ordinary one-way ANOVA, with Tukey post hoc test (0.05 cut-off).

## 3. Results

### 3.1. SARS-CoV-2 Infection Modulated the N-Degron Pathway and Increased ATE1 Enzyme Expression

To explore protein arginylation during SARS-CoV-2 infection, we performed an in silico multiomics data analysis and validated the findings in a time-course SARS-CoV-2 infection at the protein level by immunoblotting (Figure 1A). Initially, the basal levels of enzymes involved in the N-degron pathway were evaluated in different uninfected cells (Figure 1B). Enzymes involved in protein arginylation (ATE1), ubiquitination (UBR1, UBR2, UBR4, UBR5), arginine-tRNA ligase assembly (RARS2), deamidation (NTAN1), and N-terminal methionine removal (METAP1, METAP2) were identified in all uninfected cell models with no statistical difference among them. These findings indicated that enzymes involved in the protein arginylation pathway were not modulated based on the cell type or species in uninfected conditions. A total of 918 proteins (Figure 1C) with potential to be arginylated at the N-terminus (NtE, NtD, NtC, NtN, NtQ), in agreement with the UniProt sequence, were identified in uninfected cell lines and showed a similar expression pattern (Figure 1D), regardless of the organism (Green Monkey and Human).

Furthermore, we reanalyzed eight datasets covering transcriptomic and proteomic data of in vitro and in vivo SARS-CoV-2 infection of different biological systems [51,52,53,54,55,56,57] (Figure 2A). ATE1 expression was higher during infection in most of the datasets, significantly upregulated at both transcript and protein levels [51,54,55,56,57]. On the other hand, RARS1 and RARS2 protein expressions were opposite, with RARS1 being upregulated and RARS2 (mitochondrial) downregulated. UBR1, UBR2, and UBR5 ubiquitin-ligases (E3) expressions were increased in infection; however, UBR4 was regulated in a different direction at transcript (Wu et al.) and protein (Saccon et al.) levels. The expressions of proteins involved in the removal of the N-terminal methionine were variable among the different studies. However, protein expressions of the caspase family were upregulated, and especially CASP3 expression was statistically significant in four studies [51,53,54,56].

Western blot analysis was performed to measure the ATE1 level, which confirms the above findings (Figure 2B). Consistent with the omics data, Calu-3 and Vero CCL-81 cells infected with SARS-CoV-2 had statistically higher ATE1 levels compared to the uninfected CTRL group (Figure 2B). Time course data revealed an increase in ATE1 after 2 h of infection in Vero CCL-81 cells (*p*-value = 0.0294). On the other hand, statistical significance between groups was found after 48 h (*p*-value = 0.0259) in Calu-3 cells. It was found that the ubiquitin-conjugating E2 enzymes (UBE2G2, UBE2L3, UBE2D2, UBE2D3, UBE2K, UBE2D4, UBE2R2, UBA52, UBE2A, UBA3, UBE2W, UBE2L6, and UBE2E1) were also overexpressed in the infected groups (Appendix A). To verify whether transfection with the spike protein of the Wuhan variant (WT) of SARS-CoV-2, instead of the whole virus, would be able to induce modulation of protein arginylation, HEK 293T cells were evaluated after 24, 48, and 96 h of infection (Figure 2C). An increase in ATE1 was identified after 96 h of transfection compared to the CTRL empty vector group. Transient transfection with different receptor binding domains (RBD) can also modulate arginylation. We found that the DELTA variant has a more remarkable ability to induce ATE1 levels than the WT, BETA, and P1 variants after 96 h (Figure 2D).

### 3.2. Increased ATE1 Expression in SARS-CoV-2 Infection Was Correlated with Events Linked to the Endoplasmic Reticulum (ER)

Once the increased abundance of ATE1 in the infection was confirmed, a multicorrelation analysis was performed using omic data to verify which proteins correlated with ATE1 (Figure 3A), and which pathways could be associated with this increased abundance. Only differentially regulated proteins/genes were selected from six studies on SARS-CoV-2 [51,52,54,55,56,57]. A total of 365 proteins/genes presented a significant correlation (*p*-value < 0.05) with ATE1 in at least two studies and 28 in at least three studies (Appendix A). Analyzing the molecular functions (MF) of the 28 correlated proteins/genes, the enrichment of processes related to unfolded protein binding, protein-folding chaperone, and ubiquitin-protein ligase binding was found (Figure 3B). Among the biological processes (BP), events related to ER and viral infection were enriched, such as protein target to ER, protein localization to ER, viral gene expression, and viral transcription (Figure 3C). Pathways related to alterations in processes linked to RNA and coronavirus infection were also enriched (Figure 3D). The GBP2 protein, involved in the cellular response to infections, was correlated with ATE1 in four studies. Due to the observed relationship between the processes linked to the ER (Figure 3B,C), we monitored the direction of the correlation of HSPBP1 (Figure 3E) and HSP90B1 (Figure 3F) with ATE1. These proteins showed significant positive correlations, except for the negative correlation observed in lung tissue by Qiu et al. [54].

After identifying a relationship between ATE1 and ER-associated chaperones/processes during SARS-CoV-2 infection, the arginylation levels of proteins located in the ER, heat shock protein family A (Hsp70) member 5 (HSPA5, also known as BiP), calreticulin (CALR), and protein disulfide isomerase (PDI) were analyzed by Western blotting (Figure 4). The BiP/HSPA5 arginylated protein level increased in both cell models over time with statistical significance 48 h after the onset of infection (Figure 4A,B). Interestingly, the arginylated CALR protein level decreased in Calu-3 cells, while it increased in Vero CCL-81 (Figure 4B) compared to the CTRL uninfected cells. PDI protein showed a significant increase 2 h after the onset of infection in Calu-3 cells, and was statistically more arginylated in infected Vero CCL-81 cells after 48 h (Figure 4C). Arginylated proteins are targets of multiple pathways, including autophagy via binding to p62 and LC3B, as described by previous reports [34,81,82]. Based on this, we monitored and confirmed that Calu-3 and Vero CCL-81 cells present different modulation of the autophagy pathway when infected by SARS-CoV-2, especially of p92/SQSTM1 and LC3B proteins (Appendix A). Thus, the substrates (CALR, BIP, and PDI) may present different behavior due to the modulation of the autophagy pathway. Furthermore, studies have shown that arginylated proteins can be relocated in the intracellular space [83] and be more accessible or inaccessible to degradation, resulting in a differential arginylation profile according to the substrate studied.

Searching for other organelles involved in arginylation during SARS-CoV-2 infection, we performed a subcellular localization analysis of proteins correlated with ATE1 in at least two studies (Appendix A). These proteins mostly occupy complexes of chaperones, ribosomal, proteasome, cytoskeletal microtubules, and actin filament. Recently, Seo et al. [65] and Wong et al. [66] demonstrated experimentally that 152 were arginylated proteins, including mainly actins, chaperones, ribosomal components, and tubulins (Appendix A), and nine proteins (VIM, HSPB1, PRDX4, ACTG1, ACTB, CALR, ATP5F1A, SPTAN1, and HSPA1B) overlapped in both studies. Bringing together arginylated proteins that were differentially regulated during SARS-CoV-2 infection and presented the same direction of regulation (upregulated or downregulated) in at least two studies (Appendix A), we observed that tubulins and chaperones were increased in the infected group (INF); on the other hand, VIM and SPTAN1 proteins were downregulated. Collectively, data analysis of differentially regulated proteins pointed to an increased level of arginylated proteins in SARS-CoV-2 infection (Appendix A). Looking at the arginylated proteins evaluated here (Figure 4), we found that they are differentially regulated in distinct directions in the studies by Saccon et al. [51], Nie et al. [52], Wu et al. [56], and Leng et al. [53] (Appendix A). The subcellular location of the 152 arginylated proteins was mainly in the cytoskeleton, cytoplasm, and nucleus (Appendix A). Since the ACTB protein was previously identified as arginylated by Seo et al. [65] and Wong et al. [66], Western blot analysis was performed to measure arginylated ACTB levels in infected Vero CCL-81 and Calu-3 cells (Appendix A). Increased levels of ACTB arginylation were observed for up to 24 h in Calu-3 cells, with a reduction 48 h after infection. On the other hand, in Vero CCL-81 cells, the increase in arginylation occurred only after 48 h. ACTB downregulation was identified by Wu et al. in lung tissue obtained from patients who died from COVID-19 in Wuhan, China (Appendix A).

### 3.3. The Increase in ATE1 Levels Occurs Earlier with the Brazilian Variants P1 and P2 Compared to the Wuhan Variant (WT)

After verifying the modulation of arginylation resulting from infection by the Wuhan SARS-CoV-2 (WT) variant, Calu-3 cells were infected with the P1 and P2 variants, which were isolated for the first time in Brazil. In addition, 1 uM of the ER stress inducer thapsigargin was used (Figure 5). The treatment with TAG induced the levels of ATE1, being possible to observe a significant increase in P1 in relation to the CTRL, WT, and P2 groups after 48 h of infection. The use of TAG induces stress even in the CTRL group. Untreated cells (Figure 4B) show an increase or tendency to increase ATE1 only in infected groups (WT, P1, and P2). Furthermore, we verified that at 6 h (TAG-), it is possible to verify a significant increase in P2. The TAG- groups showed an increase in BiP/HSPA5 arginylation, as previously demonstrated (Figure 4A,B). However, subjecting groups to TAG treatment, the observed behavior is the opposite. On the other hand, CALR showed increased levels of arginylation in both models (TAG+ and TAG-). Taken together, these data draw attention to a variant-dependent modulation of arginylation.

### 3.4. Transfection with Spike or RBD Highlights the Potential of SARS-CoV-2 to Induce Protein Arginylation

Transfection with Spike glycoprotein or RBD from different strains increased levels of BiP/HSPA5 arginylation (Figure 6A,B). However, in the first 48 h after transfection with Spike, a decrease in BiP/HSPA5 arginylation was observed. The increase in levels occurred after 96 h (when there was an increase in ATE1). The DELTA variant RBD was able to induce R-BiP levels in the first 48 h and maintained this increase up to 120 h after infection. The CALR protein showed increased levels of arginylation in both transfections (Figure 6A,B), as well as in the whole virus infection model (Figure 4). The transfections underscore the potential of SARS-CoV-2 or its viral particles to induce protein arginylation.

### 3.5. ATE1 Inhibition and Silencing Reduces SARS-CoV-2 Viral Release in Calu-3 Cells

In view of the close relationship between arginylation and SARS-CoV-2 infection demonstrated by previous data, enzyme inhibition assays by 1 µM tannic acid and 25 µM merbromine (MER) were performed on Calu-3 cells (Figure 7). These concentrations of tannic acid and MER did not affect cell viability. Notably, cells infected before any treatment (INF-24 h) had higher ATE1 expression than uninfected cells (CTRL), and treatment with tannic acid and MER significantly decreased the level of ATE1. Expression levels of ER proteins (CALR and BiP/HSPA5) decreased similarly to ATE1, but this was less pronounced. Of note, tannic acid and MER were able to reduce viral load or prevent virus entry into the cell. Such an effect was more relevant in the inhibition with MER (Figure 7B).

To confirm the reduction in viral load due to the decrease in ATE1 levels, an ATE1 silencing assay was performed (Figure 7C). It was possible to identify an increase in the abundance of ATE1 in the INF group in relation to the CTRL. In addition, there is a reduction in ATE1 in the silenced groups, both infected and uninfected. Although ATE1 was efficiently silenced, the BiP/HSP5A protein increased arginylation levels in the INF group. On the other hand, the CALR protein showed reduced arginylation. Viral load reduction was confirmed (Figure 7D) in the silenced infected group (INF-siATE1) in relation to the infected CTRL (INF-CTRL), confirming the results obtained with the ATE1 inhibitors. This finding strongly demonstrates that the reduction in ATE1 is correlated with the reduction in viral release.

Reaffirming the previous findings, the anti-RBD antibody was used (Figure 7E) and demonstrated lower band intensity in the INF-siATE1 group compared to the INF-CTRL group.

### 3.6. Single Cell RNA-seq Data Showed That Macrophages and Epithelial Cells Express ATE1

After verifying the expression and subcellular location of proteins involved in the arginylation process, we investigated which cell types express *ATE1*. A reanalysis of single-cell RNASeq of nasopharyngeal/pharyngeal swabs samples published by Chua et al. [67] was conducted comparing the INF group consisting of critically ill patients, hospitalized for more than 20 days or who died from the progression of COVID-19 with the CTRL group of uninfected cases. A total of 17 cell clusters were identified (Figure 8A). The gene sets of clusters 4, 7, and 8 presented a significant differential expression (*p* < 0.05) between the INF and CTRL groups (Figure 8B). *ATE1* was included in clusters 4, 9, 15, and 17 (Appendix A). The top five markers in cluster 4 were *LYZ*, *SRGN*, *HLA-DPB1*, *CD74*, and *TYROBP*, all markers of macrophages (Appendix A). Cluster 4 also contained the macrophage markers *MARCO* [84], *CD163* [85], *MRC1* [86], and *MSR1* [87], which reinforces the presence of macrophages in this cluster (Appendix A). Based on these markers, the macrophage compartment was isolated in cluster 4, and the genes differentially regulated between the CTRL and INF groups were determined (Appendix A). The upregulated genes in cluster 4 were associated with interferon type I induction and signaling during SARS-CoV-2 infection, pulmonary fibrosis, proteasome degradation, and ferroptosis; the downregulated genes were related to peptide chain elongation, oxidative phosphorylation, and MHC class II complex (Appendix A). However, the expressions of genes related to the modulation of arginylation: *ATE1*, *CALR*, *ACTB*, *PDIA3*, *PDIA6*, and *PDIA4* did not show statistical significance between the groups (Appendix A); although, the *BiP/HSPA5* gene expression was increased in the INF group.

We confirmed the absence of ATE1 modulation in macrophages at the protein level by Western blotting of macrophages infected with SARS-CoV-2 (Figure 8C). These data suggested that the arginylation behavior in infected macrophages was different from that observed in Vero CCL-81 and Calu-3 cells. The inhibitors decreased ATE1 enzyme levels in macrophages at 48 h and 24 h after treatments. The expressions of ER chaperones, CALR and BIP, were significantly decreased in infected macrophages (48 h) compared to non-infected macrophages in Western blot assays (Figure 8C). Looking at differentially regulated genes between the CTRL and INF groups in clusters 9, 15, and 17 (Appendix A), we identified a statistically significant increase in ATE1 in the INF group in cluster 15, which was enriched mainly with epithelial cell markers confirming our data on Calu-3 cells (Appendix A).

### 3.7. The N-Degron Pathway Was Regulated in SARS-CoV and MERS-CoV Infections but Not in H1N1 Influenza and Respiratory Syncytial Virus (RSV) Infections

We verified whether modulation of the N-degron pathway was recurrent in other respiratory viral infections or was a specific signature of SARS-CoV-2 (Appendix A). The identification/regulation of proteins related to N-terminal methionine removal and ubiquitination was less recurrent in influenza, RSV, and human adenovirus infections. On the other hand, viruses of the *Coronaviridae* family, such as SARS-CoV and MERS-CoV, showed modulation of the proteins involved in these reactions. Convergently, ATE1 was not identified or was downregulated in in vitro models infected with H1N1 and RSV; in contrast, it was upregulated in cells infected with SARS-CoV and MERS-CoV. These data indicated an arginylation-dependent signature during infection with viruses from the *Coronaviridae* family.

## 4. Discussion

In this study, we described the modulation of the N-degron pathway and arginylation of proteins during SARS-CoV-2 infection by a combined in silico analysis of multiomic studies and data validation by Western blotting (Figure 1). We demonstrated an increase in ATE1 expression, a critical enzyme involved in arginylation, during in vitro SARS-CoV-2 infection (Figure 2). In fact, in human Calu-3 cells, a progressive increase in ATE1 expression was observed after 6 h of SARS-CoV-2 infection, when an increase in viral proteins has previously been demonstrated [55]. Interestingly, an increase in ATE1 expression was also observed in a monkey-derived cell line (Vero CCL-81), indicating that this modulation may occur independently of the species. The increase in ATE1 occurred when cells were infected with different variants, or transfected with the spike protein, or with RBDs. The P1 and P2 variants, first identified in Amazonas, Brazil [88], were considered to be of worldwide interest, and here we demonstrate that ATE1 activation occurred earlier in these variants than in WT (Wuhan). Moreover, the increase in ATE1 was demonstrated in MERS-CoV (24 h) and SARS-CoV (36 h) infections [60], but not in other respiratory virus infections such as RSV [63,89,90], and influenza [58,59], suggesting that involvement of the N-degron pathway may be a specific molecular signature for the *Coronaviridae* family. A recent study found lower levels of side chain arginylated peptides in the plasma collected from COVID-19 patients [91]. This apparent discrepancy compared to our study could be associated to the different biological systems under investigation, such as intracellular proteins in a cell line and mRNA levels of ATE1 in human broncholavage fluid compared to plasma circulating proteins. Moreover, our study focused on the N-terminal arginylation while the previous report analyzed side chain arginylation.

Previous studies by others and our group revealed an activation of the UPR pathway after 6 h of SARS-CoV-2 infection [17], corroborating ER stress enhancement [26] and UPR pathway activation during viral infection [18,19]. Based on these findings, we hypothesized that increased misfolded or unfolded proteins produced during SARS-CoV-2 infection may be tagged for degradation by arginylation in order to maintain cellular homeostasis. In fact, in silico multiomics analysis identified several ER related proteins associated with ATE1 expression (Figure 3). We showed that cells treated with the ER stress inducer, thapsigargin (TAG), kept ATE1 upregulation longer in the P1 variant, strengthening the relationship between arginylation and ER stress. As expected, BiP/HSPA5 expression increased 48 h after infection in both human and monkey cell lines. It has been shown that the viral spike glycoprotein plays a fundamental role in SARS-CoV-2 infection in the process of receptor recognition and cell membrane fusion [2], and it induced the transcriptional activation of Hsp90β member 1 and BiP/HSPA5 chaperones [14]. Increased expression of these chaperones has resulted in increased folding and processing of abundantly expressed proteins during SARS-CoV replication [20,92]. When cells were transfected with viral RBDs, we observed increased arginylation of BiP/HSPA5, especially in the DELTA variant. Recently, Zhang et al. showed that BiP/HSPA5 is an important therapeutic target in SARS-CoV-2 infection, as it can prevent viral binding and replication. In addition, four regions of the spike protein were predicted to bind BiP/HSPA5 [93]. Moreover, protein arginylation has also been induced by transient transfection of several dsDNAs, suggesting that its modulation may also be related to the detection of pathogenic dsDNA and activation of the immune system [94]. The CALR is a protein involved in the folding and maturation of glycoproteins [95,96]. CALR decrease has been described in other viral infections (influenza virus, SFV, or VSV) leading to accelerated maturation of cellular and viral glycoproteins, with a modest decrease in the folding efficiency [97]. We speculated that the progressive reduction in CALR levels in Calu-3 cells could be associated with an acceleration of coronavirus spike glycoprotein maturation. Based on previous observations, we speculate that the UPR pathway may be activated to restore ER homeostasis, and in the event of failure, apoptotic events would be induced [98]. Moreover, the virus may also hijack the host’s ubiquitination machinery [99]; however, the biological functions of this action are still unknown.

Our multiomic analysis demonstrated that the arginylation-related proteins were mainly located in the ER, which is consistent with our in vitro results of the proteins involved in the UPR pathway. Additionally, the arginylation-related proteins were also located in the cytoskeleton, a key structure in the host–pathogen interaction [100]. Cytoskeleton proteins participate throughout the viral replication cycle, as SARS-CoV-2 enters into target cells using intermediate filament proteins, sequesters microtubules to transport itself to replication/assembly sites, and promotes the polymerization of actin filaments to exit the cell [101,102]. Moreover, cytoskeleton proteins were among the experimentally arginylated proteins identified in previous studies (Seo et al. [65] and Wong et al. [66]). We monitored the arginylation levels of one major cytoskeleton protein, ACTB, and observed its increase in the first 2 h after infection in Calu-3 cells, with a decrease at 48 h, when apoptosis-related proteins were activated [17]. The ACTB arginylation pattern was different in infected Vero CCL-81 cells, as was already observed for phosphorylation [103]. This observed difference stressed the importance of using multiple cell models to assess the cellular consequences of post-translational modifications in SARS-CoV-2 infection. Although arginylation of the ACTB protein has been explored [104,105,106], it has recently been shown that the N-terminal maturation of the protein is more complex than expected. It is believed that ACTB is terminated mainly by the enzyme NAA80, which acetylates the N-terminus of exposed Asp residues, and not by ATE1 [107].

Our analysis of single cell RNA-seq data highlighted macrophages as the cells within the clusters with differential gene expression levels in infected patients. In fact, the characterization of immune cells in bronchoalveolar lavage fluid has shown that pro-inflammatory monocyte-derived macrophages were more abundant in patients with severe SARS-Cov-2 infection than in those with moderate disease or healthy individuals. Furthermore, critically ill patients presented a lower proportion of myeloid dendritic cells, plasmacytoid dendritic cells, and T cells than patients with moderate infection [108]. We also demonstrated the expression of the ATE1 protein in macrophages; however, no difference in the abundance of proteins related to arginylation was detected comparing the CTRL and INF groups. In fact, a previous study has shown that SARS-CoV-2 was capable of infecting macrophages without causing any cytopathic effect, and the virus was also capable of inducing host immunoparalysis [109]. Moreover, we also identified a cellular compartment with epithelial cell markers presenting a significant increase in *ATE1* expression, consistent with the previous observation of ATE1 protein expression in the main lung epithelial cells, ciliary cells type 1 and 2, infected by SARS-CoV-2 [110]. ATE1 expression in lung epithelial cells was higher in SARS-CoV-2 infected patients compared to controls.

Notably, we found that arginylation inhibitors, tannic acid and MER, decreased the viral load or prevented viral entry into the cell. Furthermore, our assays indicated a decrease in the abundance of ATE1. Tannic acid was recently described as a potent inhibitor of SARS-CoV-2 through the thermodynamically stable binding to the Mpro and TMPRSS2 proteins, crucial for the entry of the virus into the cell [111]. Here, we confirm the potential of tannic acid to reduce viral load, and furthermore, to modulate ATE1 levels during infection. In addition, we also demonstrated that, like tannic acid, the arginylation inhibitor MER was also capable of reducing both viral load and ATE1 level. Suramin treatment was shown to inhibit SARS-CoV-2 binding to the receptor, entry, and viral replication in Vero CCL-81 and Calu-3 cells [112]. Interestingly, suramin was also shown to inhibit ATE1 activity [113], confirming the results obtained in our study. It is important to mention that tannic acid and merbromin can modulate several intracellular signaling pathways and influence viral replication, independently of the levels of ATE1. Indeed, exposure to ATE1 inhibitors has been shown to activate/deactivate several pathways [111]. Therefore, to elucidate the direct influence of ATE1 on viral load reduction, we performed silencing of ATE1 in infected Calu-3 cells and confirmed the direct influence of the enzyme levels on viral replication, thus presenting the arginylation pathway as an important mechanism in SARS-CoV-2 infection.

## 5. Conclusions

Here, we elucidated the role of protein arginylation and modulation of the N-degron pathway during SARS-CoV-2 infection. Differential regulation of proteins involved in all reactions that make up the N-degron pathway was demonstrated, with emphasis on the upregulation of the ATE1 protein, evidenced by omics and Western blot data. Furthermore, the whole virus is able to promote this increase in ATE1 levels, but also transfection with the spike protein and RBD regions of the virus. We verified that variants declared as being of worldwide interest, P1 and DELTA, present higher levels compared to the classic variant (WT). We also showed that proteins that have their levels correlated with ATE1 perform biological functions linked to chaperone activity and binding to unfolded proteins. In fact, the ER stress inducer, TAG, increased ATE1 levels. Importantly, our findings revealed that modulation of the N-degron pathway differs between different types of infected cells, such as macrophages, Vero CCL-81, HEK 293T, and Calu-3 cells. Finally, we showed the importance of this process by reducing viral load using tannic acid and MER, known arginylation inhibitors. To strongly evidence the relationship between arginylation and SARS-CoV-2 infection, we showed that ATE1 silencing induces viral load reduction.

## Figures and Tables

**Figure 1 viruses-15-00290-f001:**
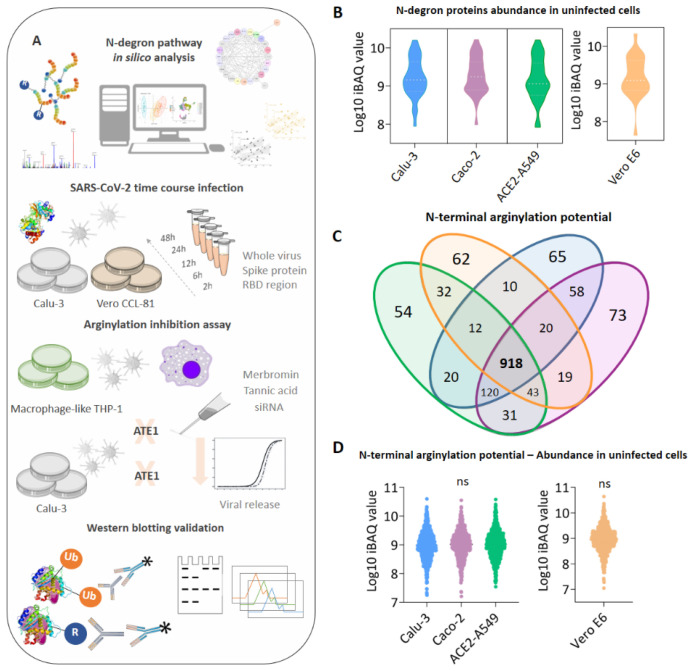
N-degron pathway modulation in uninfected cell models. (**A**) Experimental workflow adopted to identify the modulation of protein arginylation during SARS-CoV-2 infection; (**B**) expression profile of enzymes participating in the N-degron pathway identified in uninfected Calu-3 (blue), Caco-2 (purple), ACE2-A549 (green), and Vero E6 (orange) cells; (**C**) proteins with potential to be arginylated at N-terminal (NtE, NtD, NtC, NtN, NtQ) identified and quantified in uninfected cell models; (**D**) expression profile of the 918 proteins with potential to be arginylated identified in uninfected Calu-3, Caco-2, ACE2-A549, and Vero E6 cells. Proteins with the potential to be arginylated were determined based on their sequences deposited in Uniprot (*Homo sapiens* and *Chlorocebus sabaeus*). The symbol “ns” indicates a non-significant statistical relationship between groups.

**Figure 2 viruses-15-00290-f002:**
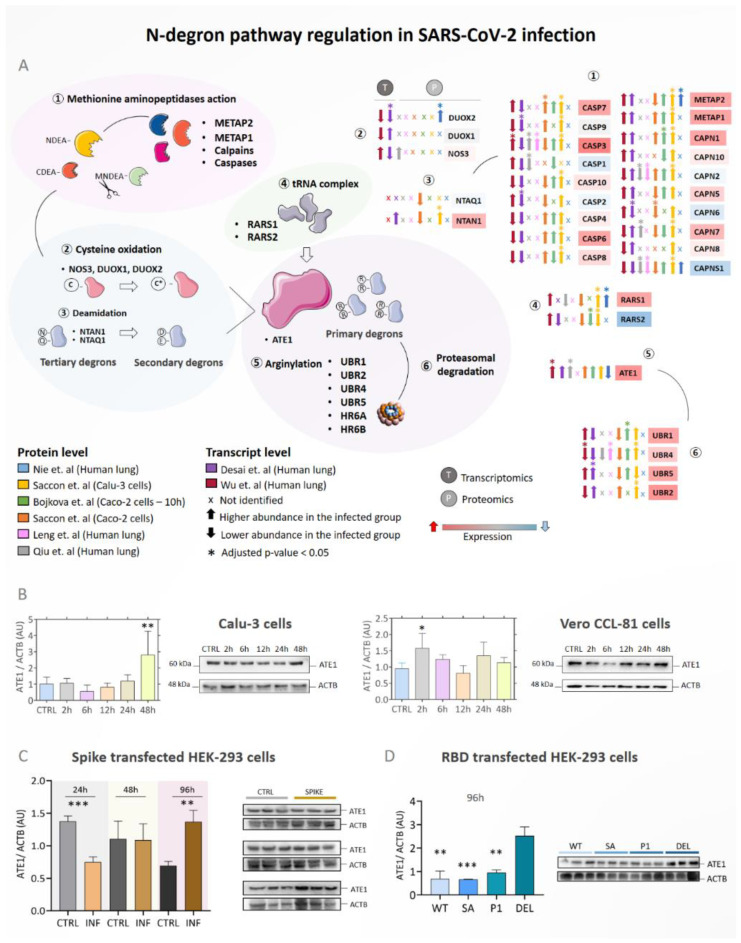
N-degron pathway modulation in infected in vitro and in vivo models. (**A**) Regulation of proteins involved in the N-degron pathway during SARS-Cov-2 infection. Proteins/genes were considered differentially regulated if they had a *q*-value < 0.05 (Benjamini–Hochberg) and were indicated by the symbol (*). The up arrows indicate proteins/genes with a higher abundance in the infected group (INF), and the down arrows indicate a lower abundance in the infected group. The color of the boxes of proteins/genes indicates the fold change (INF/CTRL) considering all the studies evaluated; the red color indicates higher abundance in the infected group and the blue color lower abundance in the INF group. The symbol (x) indicates that a protein/gene was not identified in the dataset; (**B**) Western blotting analysis of ATE1 protein in Calu-3 and Vero CCL-81 cells (n = 3) infected with SARS-CoV-2 (Wuhan strain) after 2 h, 6 h, 12 h, 24 h, and 48 h; (**C**) HEK 293T cells transfected with the SARS-CoV-2 (classical variant) spike protein; (**D**) modulation of ATE1 resulting from transfection in HEK 293T cells with the receptor-binding domains (RBD) of the WT (Wuhan), BETA (South Africa), P1 (Brazil), and DELTA (India) variants. Each point represents an independent experiment (*n* = 3). The level of significance indicates *** *p* < 0.001; ** *p* < 0.005 in relation to the control group (CTRL).

**Figure 3 viruses-15-00290-f003:**
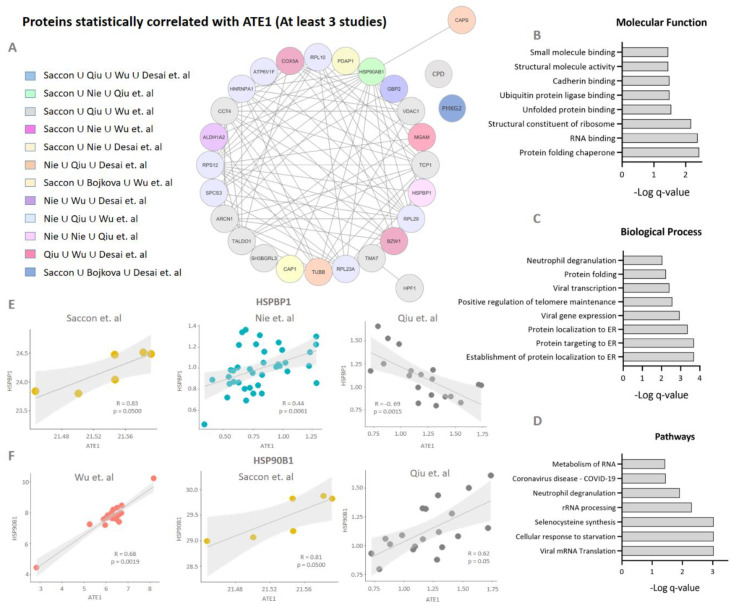
Multi-correlation expression analysis. (**A**) Proteins and genes correlated with ATE1 expression in at least three reanalyzed studies. The correlation analysis was determined by applying the Spearman test with a cut-off significance of *p*-value < 0.05. Only differentially regulated proteins/genes were considered for the correlation analysis; (**B**) gene ontology (GO) analysis of molecular functions; (**C**) biological processes and (**D**) pathways related to proteins/genes correlated with ATE1 in at least three studies; (**E**) correlation graph of HSPBP1 and (**F**) HSP90AB1 proteins indicates the positive/negative correlation with ATE1.

**Figure 4 viruses-15-00290-f004:**
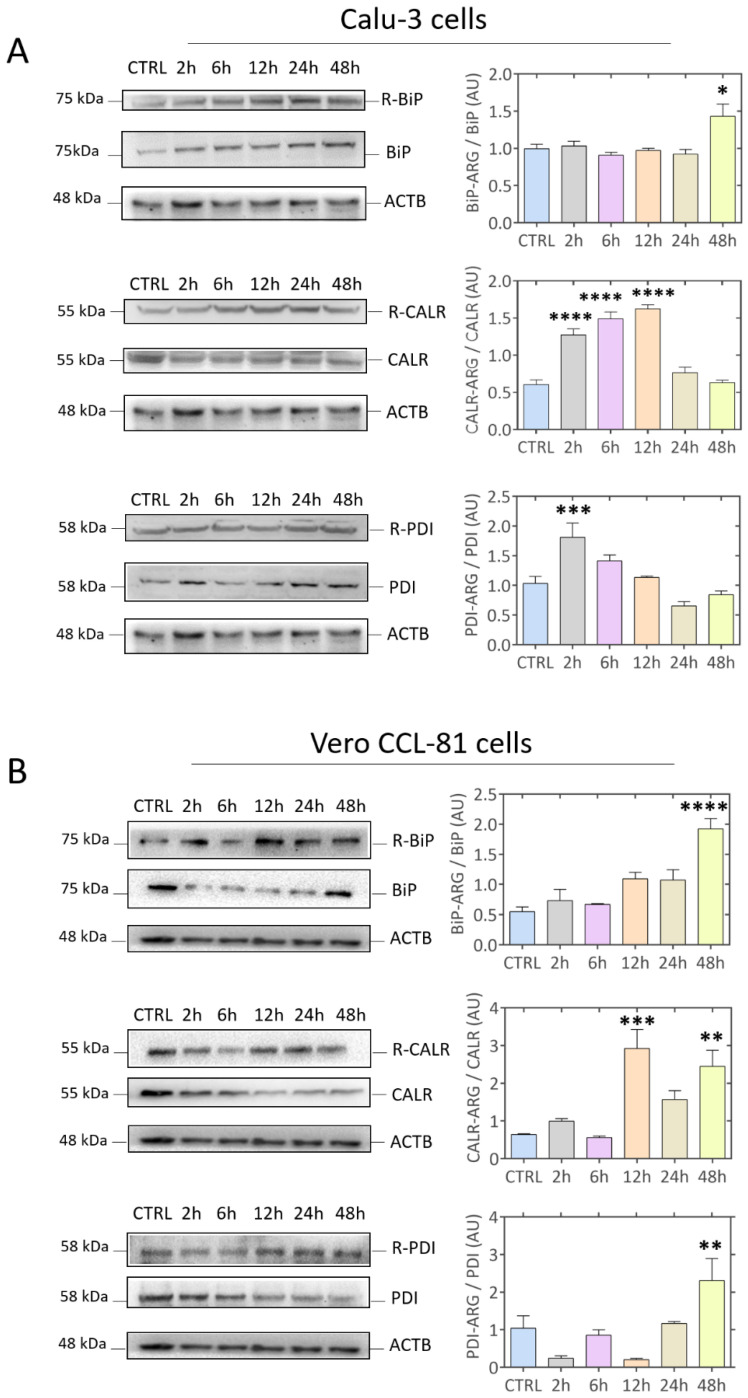
Modulation of arginylated proteins located in the endoplasmic reticulum (ER). (**A**) Representative Western blot images of R-BiP/BiP, R-CALR/CALR, and R-PDI/PDI proteins in Calu-3 and (**B**) Vero CCL-81 cells after 2 h, 6 h, 12 h, 24 h, and 48 h of infection (Wuhan strain). Each point represents an independent experiment (*n* = 3). The level of significance indicates: **** *p* < 0.0001; *** *p* < 0.001; ** *p* < 0.005; * *p* < 0.05 in relation to the control group (CTRL).

**Figure 5 viruses-15-00290-f005:**
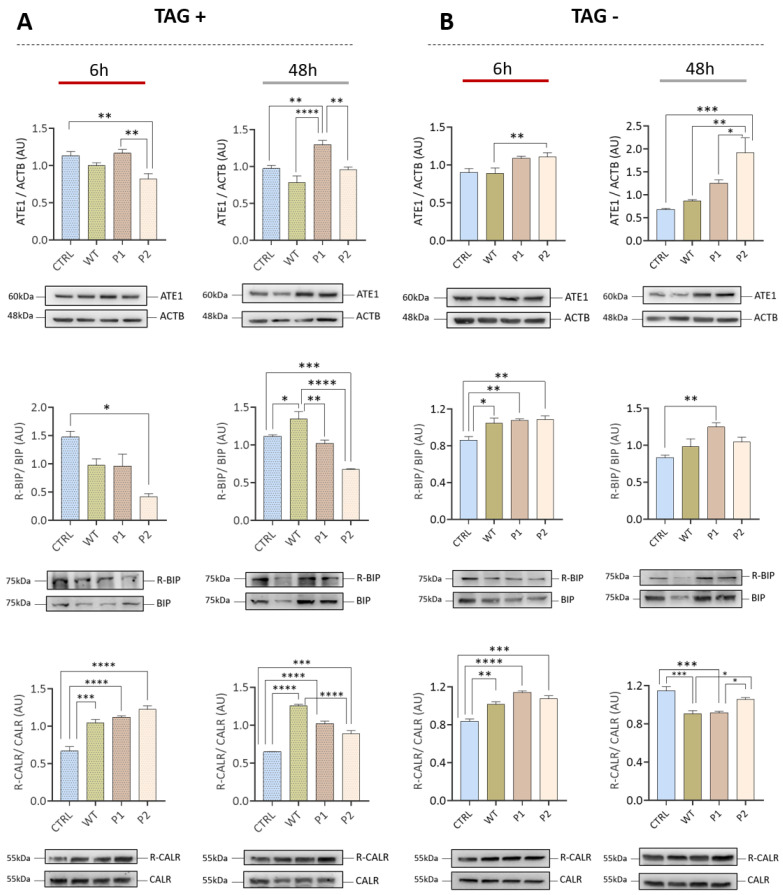
Modulation of arginylation after infection with WT (Wuhan), P1, and P2 (Brazil) variants. Calu-3 cells infected with WT, P1, and P2 variants were evaluated at 6 hpi and 48 hpi when exposed to the stress inducer thapsigargin (**A**) or not exposed (**B**). Each point represents an independent experiment (*n* = 3). The significance level indicates: **** *p* < 0.0001; *** *p* < 0.001; ** *p* < 0.005; * *p* < 0.05.

**Figure 6 viruses-15-00290-f006:**
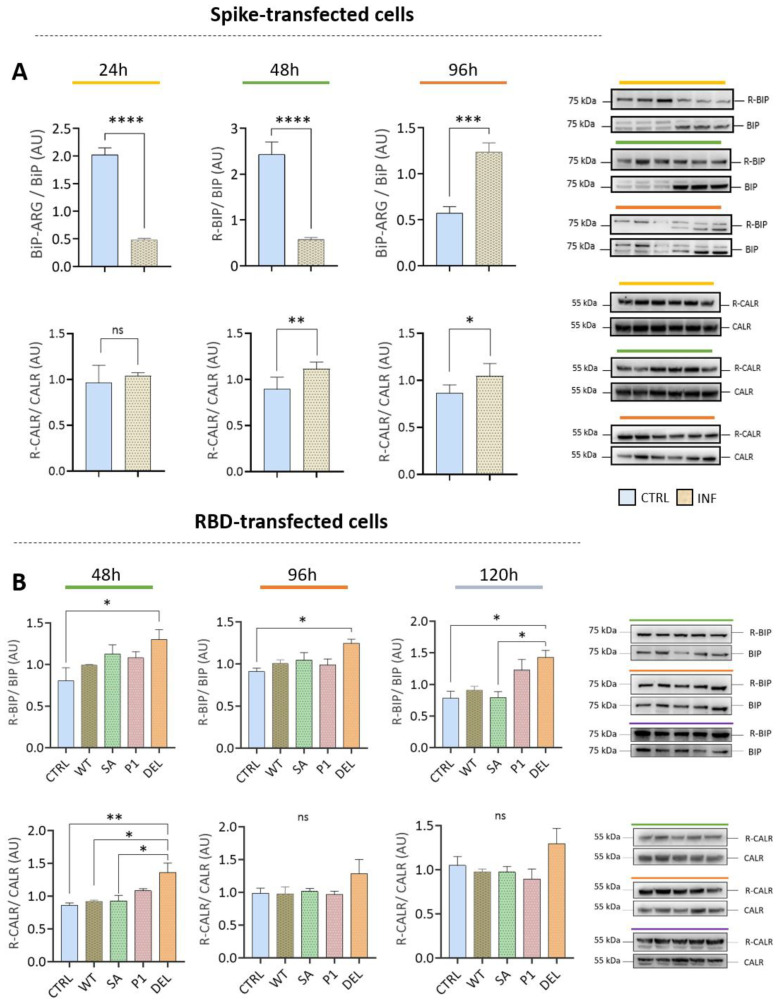
HEK 293T cells transfected with the SARS-CoV-2 (classical variant) spike protein and the receptor-binding domains (RBD). (**A**) Representative images of Western blot analysis of R-BiP/BiP and R-CALR/CALR proteins of cells transfected with the spike protein. (**B**) Representative images of Western blot analysis of R-BiP/BiP and R-CALR/CALR proteins of cells transfected with the RBD region of the wild-type (classic), Beta (South Africa), P1 (Brazil), and Delta (India) variants. Each point represents an independent experiment (*n* = 3). The level of significance indicates: **** *p* < 0.0001; *** *p* < 0.001; ** *p* < 0.005; * *p* < 0.05 in relation to the control group (CTRL). Each point represents an independent experiment (*n* = 3). ns indicates no significance.

**Figure 7 viruses-15-00290-f007:**
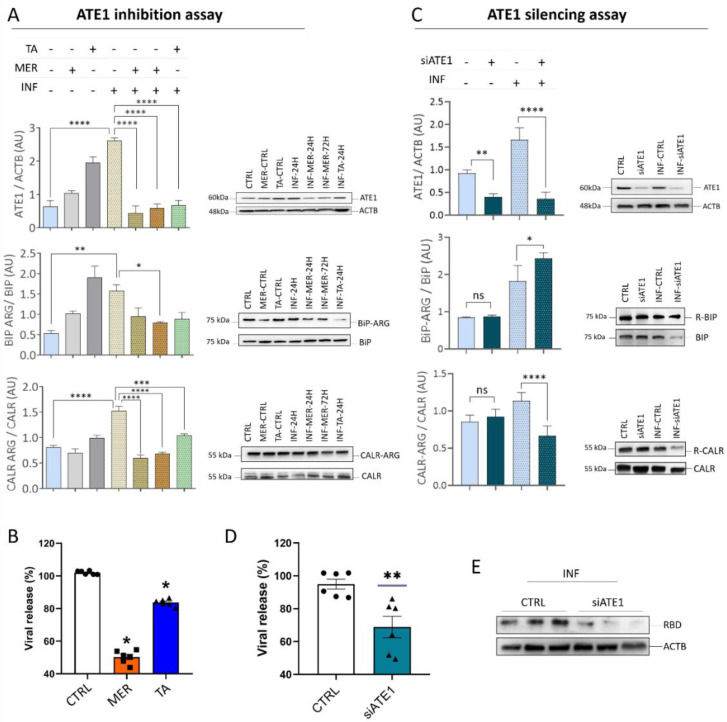
ATE1 inhibition and silencing assay in Calu-3 infected cells. (**A**) Representative images of the Western blot analysis performed on cells exposed to the inhibitors merbromin (MER, 25 µM) and tannic acid (TA, 1 µM) for ATE1/ACTB, R-BiP/BiP, R-CALR/CALR, and R-PDI/PDI proteins; (**B**) PCR results for viral release after exposure to the inhibitors; (**C**) representative images of the Western blot analysis performed on Calu-3 cells silenced for *ATE1*. The proteins ATE1/ACTB, R-BiP/BiP, and R-CALR/CALR were evaluated; (**D**) PCR results for viral release after *ATE1* silencing; (**E**) Western blotting of infected Calu-3 cells for INF-CTRL and INF-siATE1 groups using anti-RBD antibody. Each point represents an independent experiment (*n* = 3). The level of significance indicates: **** *p* <0.0001; *** *p* < 0.001; ** *p* < 0.005; * *p* < 0.05. ns indicates no significance.

**Figure 8 viruses-15-00290-f008:**
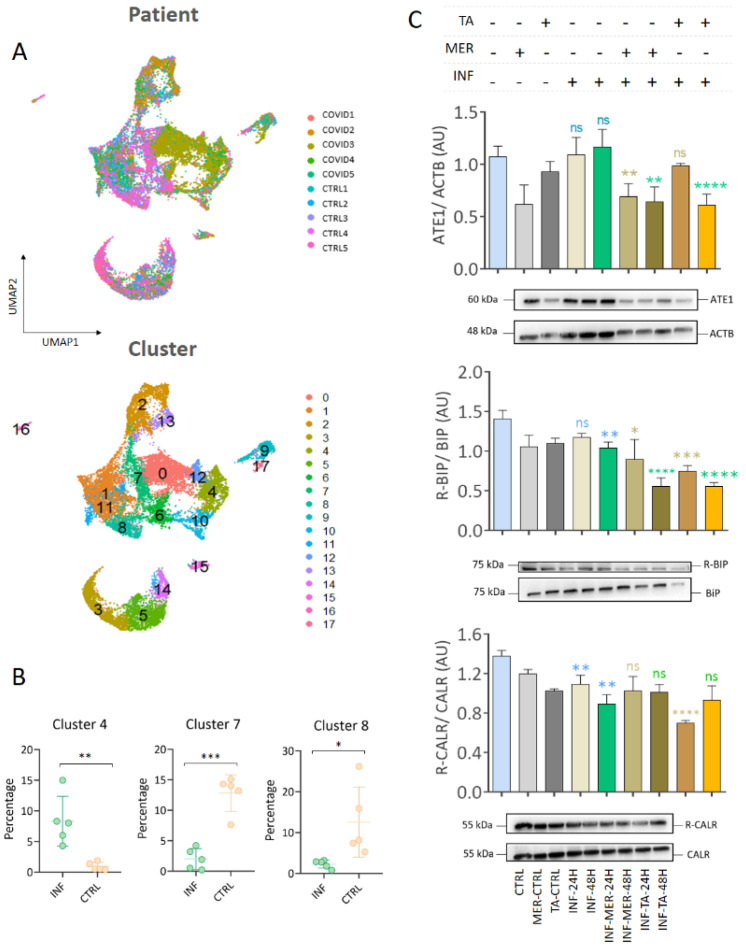
Single-cell RNA-seq analysis of nasopharyngeal samples. (**A**) Single-cell RNA-seq analysis indicating cell clustering by patients and by cell type; (**B**) differentially regulated clusters (*p*-value < 0.05) between the infected (INF) and control (CTRL) groups; (**C**) representative images of Western blotting analysis of infected macrophages, indicating the modulation of ATE1 and arginylated and total CALR and BiP proteins. Each point represents an independent experiment (*n* = 3). The level of significance indicates: **** *p* < 0.0001; *** *p* < 0.001; ** *p* < 0.005; * *p* < 0.05. ns indicates no significance.

**Table 1 viruses-15-00290-t001:** Primary and secondary antibodies used in Western blotting analyses, with their respective dilutions, reference catalog number, type, and supplier company.

Antibody	Secondary Antibody	Dilution	Reference	Type	Company
Anti-ATE1	Goat Anti-Rat	1:1000	MABS436	Monoclonal	Merck Millipore
Anti-R-BIP	Goat Anti-Rabbit	1:1000	ABS2103	Polyclonal	Merck Millipore
Anti-R-PDI	Goat Anti-Rabbit	1:1000	ABS1655	Polyclonal	Merck Millipore
Anti-R-CALR	Goat Anti-Rabbit	1:1000	ABS1671	Polyclonal	Merck Millipore
Anti-R-ACTB	Goat Anti-Rabbit	1:1000	ABT264	Polyclonal	Merck Millipore
Anti-BIP	Goat Anti-Rabbit	1:1000	#3183	Polyclonal	Cell Signaling
Anti-PDI	Goat Anti-Mouse	1:1000	#MA3-019	Monoclonal	Sigma-Aldrich
Anti-CALR	Goat Anti-Rabbit	1:1000	ab2907	Polyclonal	Abcam
Anti-ACTB	Goat Anti-Mouse	1:10,000	#A2228	Monoclonal	Sigma-Aldrich
Anti-GAPDH	Goat Anti-Mouse	1:500	sc-137179	Monoclonal	Santa Cruz Biotechnology
Anti-UB	Goat Anti-Mouse	1:1000	sc-8017	Monoclonal	Santa Cruz Biotechnology
Anti-p62	Goat Anti-Rabbit	1:1000	g # PA5-27247	Polyclonal	ThermoFisher
Anti-LC3BII	Goat Anti-Rabbit	1:1000	# PA1-16930	Polyclonal	ThermoFisher
Goat Anti-Rat	-	1:1000	BA-9400	Polyclonal	Vector Laboratories
Goat Anti-Rabbit	-	1:4000	ab6721	Polyclonal	Abcam
Goat Anti-Mouse	-	1:4000	ab6789	Polyclonal	Abcam

## Data Availability

The datasets generated during and/or analyzed during the current study are available in the public repositories as described in the Materials and Methods section in the paragraph entitled: Data sources and curation.

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
