# Peer review of "Protein Arginylation Is Regulated during SARS-CoV-2 Infection"

_viruses, 2023, doi:10.3390/v15020290_

Round 1

Reviewer 1 Report

Dear Editor,

Macedo-da-Silva et. al. have presented a comprehensive study on the role of protein arginylation during SARS-CoV-2 infection. Using various cell lines like Vero CCL-81, THP1, calu 3 cells , the authors examine protein arginylation profile of proteins as well as ATE1. They also show increased expression of ATE1 after SARS-CoV-2 infection demonstrating the importance of arginylation during infection.

The authors have presented the study in a detailed and clear manner. I do however have some comments on the data that perhaps the authors could address.

1.     Abstract: Spell check Vero CCL -81 cells.

2.     Figure 1D: Please indicate that that no statistical significance was observed.

3.     Figure 2: Which pathway do DUOX1, DUOX2 appear in 2A? Please indicate. Do the authors see differences in ATE1 mRNA level after infection in Calu-3 and Vero CCL-81 cells

4.     Figure 2 and 4, 7, 8: Do the authors always observe variability in GAPDH levels between samples? In Figure 4: Are the quantifications ( For eg: BIP-Arg/BIP) normalized to GAPDH?

Reviewer 2 Report

Remarks to the Author:

SARS-CoV-2 infection induces protein arginylation and modulation of the N-degron pathway by up-regulating ATE1, which is dominated by Spike protein. This phenomenon differs between different types of cells. The protein arginylation induced by viruses is not novel. No direct evidence of the protein being arginated was seen in the present study. Through this paper, the bioinformation results are clear and consistent with reported results. However, biochemical results are not good enough.

My major comments:

Fig 2B. It seems like no significantly increases of ATE1 expression after virus infection. The number of cells varies through GAPDH bands

Fig 4. The quality of WB should be improved. There is no discernible trend

Fig 5. Many papers reported that SARS-Cov2 can induce autophagy. These data cannot explain arginylated proteins target autophagy.

Fig 7. You’d better screen all viral proteins of SARS-Cov2 to identify the role of spike on the expression of ATE1.

Fig 8. What’s the protein level of SARS-CoV2? Detect the N or E expression after SARS-CoV2 infection treated by ATE1 inhibition.

Fig 10B. GAPDH level has the same trend with ATE1, R-BIP? Why?

Round 2

Reviewer 2 Report

  • The revised article should be accepted